# Mechanisms of Separation and Crystal Growth of Mullite Grains during Preparation of Mullite-Based Ceramics from High Alumina Coal Fly Ash

**Jianbo Zhang [1,*], Huiquan Li [1,2] and Shaopeng Li [1]**

[1] National Engineering Research Center of Green Recycling for Strategic Metal Resources, Institute of Process Engineering, Chinese Academy of Sciences, Beijing 100190, China
[2] University of Chinese Academy of Sciences, Beijing 100049, China
[*] Correspondence: zhangjianbo@ipe.ac.cn; Tel.: +86-10-8254-4825; Fax: +86-10-8254-4830

**Abstract:** High-alumina coal fly ash (HAFA: 45% $Al_2O_3$ and 40% $SiO_2$) is regarded as a special solid waste that is generated from power plants in northwestern China. It is regarded as an important substitute for bauxite and is applied to prepare mullite-based ceramics. In this work, a hydrometallurgy–pyrometallurgy synergistic process is proposed to resolve the lower $Al_2O_3/SiO_2$ mass ratio (Al/Si) and lower degree of crystallinity that can promote the formation of compact mullite-based ceramics. During the activation–desilication process, the inert Al-O-Si is activated to form more active Si-O-H in acid activation, which can be decomposed easily in the desilication process, and the Al/Si mass ratio increases from 1.17 to 2.80, so the mullite grains and metastable phase can be exposed. During the sintering process, mullite grains and the metastable phase tended to axial growth, part of the metastable amorphous $Q^4(3,2,1Al)$ structure was transformed to $Q^4(4Al)$ structure (mullite), and then the staggered spatial structure was formed such that the bulk density and apparent porosity of the mullite-based ceramic reached 2.85 $g/cm^3$ and 0.5%, respectively. This process not only consumes more HAFA but also helps alleviate the shortage of bauxite, which will promote the development of clean coal-fired power generation.

**Keywords:** HAFA; acid activation; deep desilication; A/S mass ratio; mullite



## 1. Introduction

Ceramics have become a necessity in daily life, industry, and art. Billions of ceramic products, which are mainly prepared by kaolin, bauxite, corundum, and quartz, are consumed annually [1,2]. High-alumina coal fly ash (HAFA), which contains more than 40–50% $Al_2O_3$ and more than 35–45% $SiO_2$, is regarded as an important substitute. Silicon-aluminum glass, mullite, and corundum phases are mainly contained in HAFA [3–5]. About 30 million tons HAFA are discharged annually and are a cause of serious pollution [6–9]. At present, HAFA is mainly utilized for building materials [10–12] and alumina extraction [13–15], which consume large amounts of HAFA. HAFA is also used for Al–Si materials according to the properties of its components and phases [16–18]. However, the low Al/Si and poor crystallinity of mullite cannot meet the demand for mullite-based ceramics [19,20]. Therefore, elevating the Al/Si content and improving the crystallinity/structure of mullite are necessary.

Mullite is regarded as an important fire-resistant mineral (A/S = 2.55), but the A/S of HAFA is only approximately 1.0. The addition of $Al_2O_3$ [21,22] and desilication technology [23–25] are the two methods to elevate A/S. Li [16] and Foo [26] improved the A/S of HAFA to above 2.55 by adding alumina and bauxite. The mixtures are shaped and sintered at high temperature, which promotes the formation of the mullite phase and results in the good properties of a dense or porous mullite. Pre-desilication of HAFA can efficiently elevate the A/S. Guo [27] prepared mullite from original HAFA and desilicated

HAFA (DHAFA) under 1600 °C. Mullite ceramics from DHAFA exhibit better physical and mechanical properties. On this basis, Luo [28] adopted pre-desilication to elevate the A/S to 2.49, and the DHAFA was sintered at 1300 °C by using a fluxing agent (zeolite), which can exhibit optimal sintering properties. However, the A/S of the above DHAFA could not be elevated to 2.55, which could decrease its refractoriness and strength. In terms of this problem, Zhang [29] investigated the activation mechanisms of inert amorphous aluminosilicate, and it was indicated that the inert $Q^4$(3Al) structure and side reaction are the main factors that hinder deep desilication.

Based on the previous researches, a hydrometallurgy–pyrometallurgy synergistic method was studied. The influences of different factors (moisture content, forming pressure, sintering temperature, and time) on the bulk density, apparent porosity, phases, and morphology were studied and optimized, providing for the novel yet comprehensive utilization of HAFA.

## 2. Material and Methods

### 2.1. Material

Analytical regents were NaOH (>96 wt.%) and hydrochloric acid (36 wt.%–38 wt.%).

HAFA generated from pulverized coal ash was collected from in a coal-fired power plant in Inner Mongolia, China.

### 2.2. Experiments

The experimental process is shown in Figure 1.

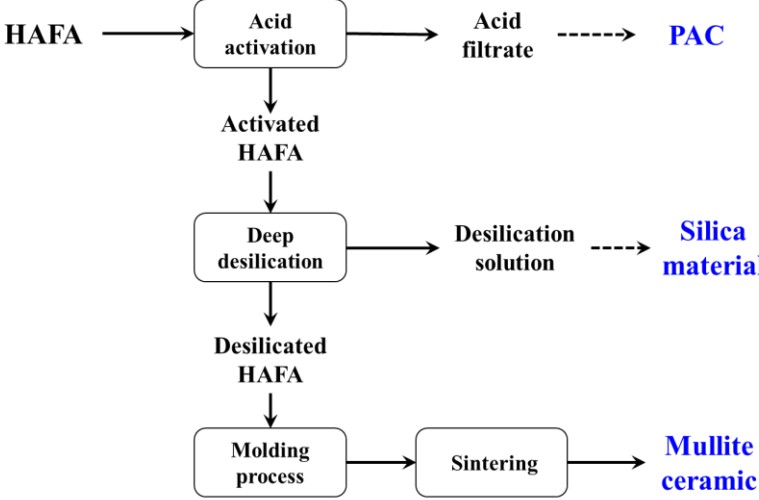

**Figure 1.** The whole experimental process.

**Acid activation:** HAFA was firstly mixed with hydrochloric acid under optimal conditions: T = 80 °C, t = 90 min, L/S = 5, C = 6 mol/L. Acid-activated HAFA (AHAFA) was washed with 80 °C deionized water at L/S = 1.5. Acid filtrate is used to prepare polymeric aluminium (PAC).

**Deep desilication:** AHAFA and an alkaline solution were mixed under the following conditions: T = 95 °C, C = 5.5 mol/L, L/S = 5, t = 90 min. After desilication, the desilicated HAFA (DHAFA) was washed with 80 °C deionized water at L/S = 1.5.

**Sintering process:** The DHAFA was mixed with additive and water under optimal pressure, and the mixture formed a green solid, which was prepared for the sintering process. A high-quality product could be obtained after optimization of the sintering process (calcination temperature and time).

### 2.3. Characterization

The elemental content was analyzed by X-ray fluorescence spectrometry (XRF) (AXIOS-MAX, 50 kV, 60 mA). The concentrations of different ions were determined using an ICP 6300 inductively coupled plasma optical emission spectrometer (Thermo Fisher Scientific, Waltham, MA, USA) in a wavelength range from 165 nm to 782 nm (±0.1 nm). The phases of different samples were analyzed by XRD (X-ray diffractometer, Empyrean, CuKα, 40 kV, 40 mA). The morphology of the samples was detected by SEM, (FEI MLA Quant 250), and the compositions of different regions on the sample were analyzed by energy-dispersive X-ray spectroscopy (EDS, EDAX). The Al-O-Si coordination structure was analyzed by solid-state $^{29}$Si MAS NMR.

## 3. Results and Discussion

### 3.1. Properties of HAFA

The chemical composition, physical properties, and mineral phases of HAFA are shown in Table 1 and Figure 2, respectively.

**Table 1.** Chemical composition of original HAFA (wt.%).

| Sample | $Al_2O_3$ | CaO | $SiO_2$ | $Fe_2O_3$ | $Na_2O$ | $TiO_2$ | MgO | Al/Si |
|--------|-----------|------|---------|-----------|---------|---------|------|-------|
| HAFA | 45.87 | 1.85 | 39.37 | 1.60 | 0.33 | 1.62 | 0.2 | 1.17 |

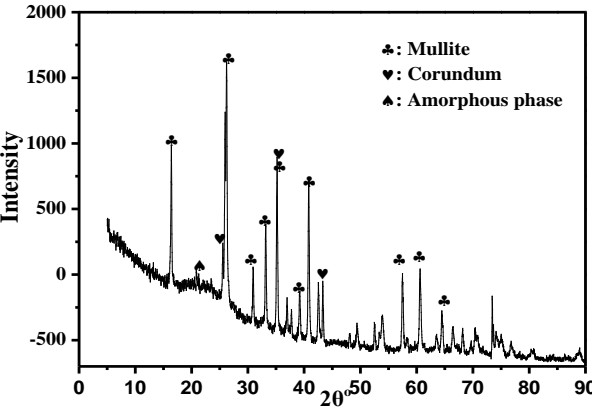

**Figure 2.** XRD pattern of high alumina fly ash.

The chemical composition of original HAFA is shown in Table 1. It indicated that the combined $Al_2O_3$ and $SiO_2$ content exceeded 85%, and the Al/Si mass ratio was 1.17. The other impurities included $Fe_2O_3$, CaO, $TiO_2$, C, etc.

The XRD pattern (Figure 2) and SEM (Figure 3) together showed that HAFA contains three phases: mullite, corundum and amorphous aluminosilicate. The bulge in the XRD pattern indicated an amorphous phase (ranging from 10° to 30°), which was caused by the quenching of HAFA under approximately 1300 °C. At the same time, many mullite grains were formed and were easily wrapped by amorphous-phase materials.

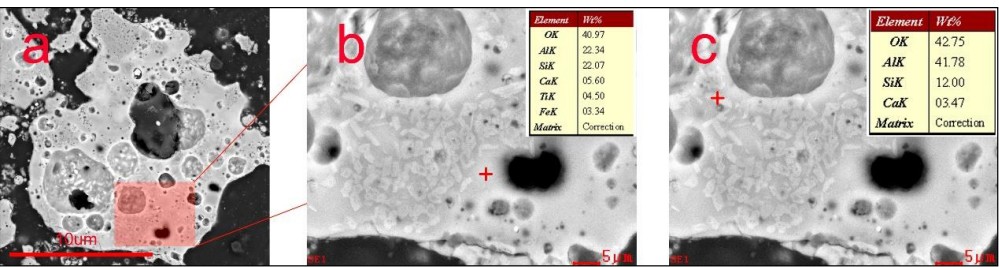

**Figure 3.** SEM images of HAFA (**a**) and EDX analysis of amorphous phase (**b**) and mullite phase (**c**).

Figure 4 shows that the amorphous silicate was in the form of $Q^4(3Al)/Q^4(2Al)/Q^4(1Al)$ coordination structure beside the Q4(0Al), and the Al-O- coordination decreased the reactivity of the Si-O- of the amorphous aluminosilicate in the dilute alkaline solution. Therefore, it was necessary to activate HAFA by removing the Al in the amorphous silicate.

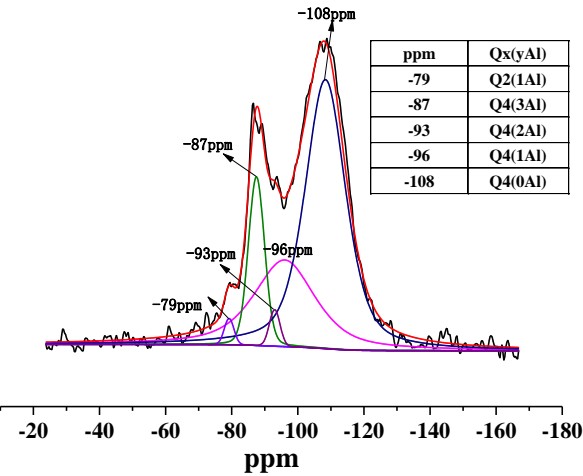

| ppm | Qx(yAl) |
|-----|---------|
| -79 | Q2(1Al) |
| -87 | Q4(3Al) |
| -93 | Q4(2Al) |
| -96 | Q4(1Al) |
| -108 | Q4(0Al) |

**Figure 4.** $^{29}$Si MAS NMR spectra of HAFA.

### 3.2. Investigation of Activation–Deep Desilication Process

Based on previous studies, an acid activation–deep desilication process was adopted, and the active Al in the amorphous phase was decomposed and replaced by H$^+$, which helped to form an active OH site, and the amorphous phase became flocculent and disordered, as seen in Figure 5a,b. This process can elevate the reactivity of Si-O- in amorphous aluminosilicate. After he process, it was observed that the Al/Si elements were enriched in the same area (Figure 6b), and the flocculent amorphous phase was decomposed, while the corundum and mullite phases were exposed, and the needle-like crystal phase (mullite) was exposed (Figures 5c and 6a).

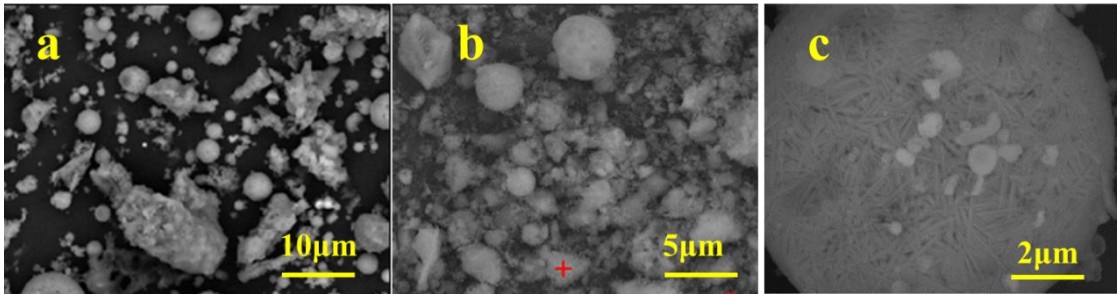

**Figure 5.** Morphologies of HAFA (**a**), AHAFA (**b**) and DHAFA (**c**).

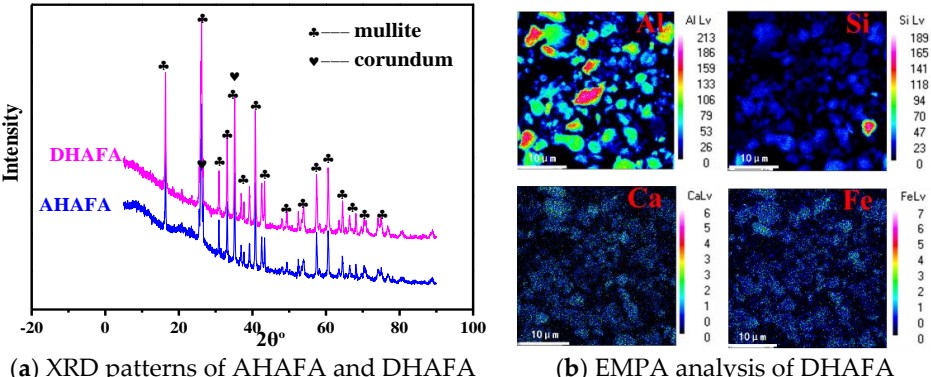

(**a**) XRD patterns of AHAFA and DHAFA        (**b**) EMPA analysis of DHAFA

**Figure 6.** XRD (**a**) and EMPA (**b**) analysis of different treated HAFA samples.

From Table 2, it can be seen that this process increased the $Al_2O_3$ content to 67.09%, but the $SiO_2$ content was decreased to 22.94%, and the Al/Si ratio was elevated to 2.80, which exceeded the Al/Si mass ratio of main-phase mullite (3 $Al_2O_3$ 2 $SiO_2$). The desilicated HAFA is regarded as a substitute raw material for mullite ceramics.

**Table 2.** Chemical compositions of AHAFA and DHAFA (wt.%).

| Sample | SiO$_2$ | Al$_2$O$_3$ | CaO | Fe$_2$O$_3$ | TiO$_2$ | MgO | Na$_2$O | Al/Si |
|---|---|---|---|---|---|---|---|---|
| AHAFA | 44.43 | 45.16 | 0.21 | 0.64 | 1.42 | 0 | 0 | 1.02 |
| DHAFA | 22.94 | 67.09 | 0.12 | 0.73 | 1.44 | 0 | 0 | 2.80 |

### 3.3. Optimization and Characterization of Mullite Properties

3.3.1. Effects of Forming Process on Mullite Properties

DHAFA has poor plasticity, and the shaping pressure and moisture content in the forming process have a positive impact on the sintering process, which can affect the mullite properties. The sintering of fine mullite seeds in DHAFA is a solid–solid reaction. Thus, the properties of the product can be easily affected by the densification of the original DHAFA.

Moisture content can change the HAFA plasticity, which can help the forming process. High shaping pressure can elevate the densification, which promotes the sintering process. The effects of shaping pressure and moisture content on the properties of mullite are shown in Figure 7.

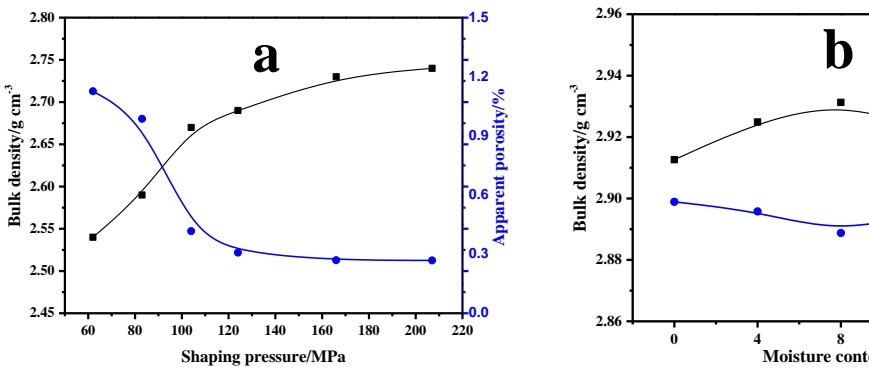

**Figure 7.** Effects of shaping pressure (**a**) and moisture content (**b**) on the mullite properties.

The influences of compressive pressures on the properties of the mullite (under the following conditions: moisture content = 10 %, sintering temperature = 1600 °C and t = 2 h) are shown in Figure 7a. With the increase in pressure from 60 MPa to 168 MPa, the bulk density of mullite changed from 2.35 g/cm$^3$ to 2.73 g/cm$^3$, but the apparent porosity of the mullite changed from 1.28% to 0.27%. The densification of particles can promote a better sintering process. With the increase in forming pressure from 168 MPa to 210 MPa, the properties of the mullite product were invariant, because the densification property had achieved a terminal level. Therefore, 168 MPa was selected as the optimal forming pressure.

The impacts of moisture content on the properties of mullite (under the following conditions: shaping pressure = 168 MPa, sintering temperature= 1600 °C and t = 2 h) are shown in Figure 7b. Increasing the water percentage from 0% to 8% increased the bulk density from 2.91 g/cm$^3$ to 2.93 g/cm$^3$, but decreased the apparent porosity from 0.39% to 0.29%. However, these properties decreased when the water percentage was above 8% because of the evaporation of excess water, which damaged the stable compact structure. Hence, basically few effects on the properties of product were observed, and the bulk density was above 2.91 g/cm$^3$, while the porosity of mullite was below 0.4%. Therefore, 8% moisture content was selected to avoid pulverization and promote better growth of fine mullite seeds.

### 3.3.2. Effects of Calcination Process on the Properties of Mullite-Based Ceramics

Calcination temperature is a key parameter in the sintering process. The fine mullite seeds cannot grow and combine at low calcination temperatures, but will aggregate and shrink because of fusion with small amounts of amorphous aluminosilicate at high temperatures. The product properties are closely related to the microstructure of mullite crystals, which mainly depends on control of the calcination temperature. Therefore, the effect of temperature on the product must be investigated.

The influences of calcination temperatures on the bulk density and porosity of mullite were investigated under optimal pretreatment conditions for 2 h. The results are shown in Figure 8. With the increase in temperature from 1400 °C to 1650 °C, the bulk density changed from 2.36 g/cm³ to 2.94 g/cm³, while the apparent porosity changed from 22.90% to 0.46%. In this temperature region, the fine mullite seeds aggregated to form a stable compact structure, which enhanced the compressive and rupture strength. The bulk density was maintained at 2.94 g/cm³, and the apparent porosity of the mullite product was kept at 0.5% with the increase in temperature from 1650 °C to 1700 °C. Therefore, the optimal calcination temperature of 1650 °C was selected to decrease the consumption of energy.

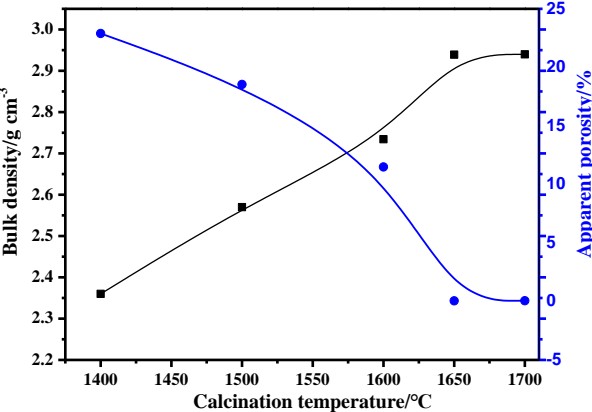

**Figure 8.** Influence of calcination temperature on properties of mullite.

The morphology of mullite under different calcination temperatures is presented in Figure 9. The size of the fine mullite seeds that aggregated together was only dozens of nanometers. While temperature was increased to 1450 °C, the accumulated mullite seeds began to grow, and the size of the crystal increased to several microns. The accumulated small crystals separated gradually when the temperature was 1550 °C. The crystals were basically separated with the increase in temperature to 1650 °C; the sizes of the crystals were maintained in the range of 5–10 μm, and they formed a stable skeleton structure that guaranteed the strength of the product.

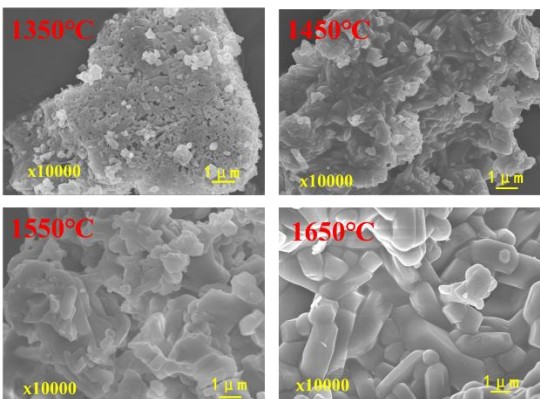

**Figure 9.** Morphology of the mullite under different calcination temperatures.

Calcination time is another key parameter during the sintering process. Traditionally, fine mullite seeds can be crystalized entirely with a prolonged sintering time. HAFA is generated from coal combustion under high temperature, and numerous mullite seeds are formed, which can accelerate the combination and growth of mullite seeds. To improve the product properties and reduce the consumption of energy, the optimization of sintering time is crucial.

The influences of different calcination times on the properties of mullite under optimal conditions are presented in Figure 10. Increasing the calcination increases bulk density but decreases apparent porosity. With the increase in calcination time from 1 h to 3 h, the bulk density of the mullite changed from 2.75 g/cm$^3$ to 2.87 g/cm$^3$, and its apparent porosity changed from 10.78% to 0.49%. The fine mullite seeds will accumulate to form a compact structure with increased calcination time, which can help improve the product strength. However, when the calcination time exceeds 3 h, the bulk density is maintained at 2.87 g/cm$^3$, while the apparent porosity is maintained at 0.5%. Therefore, an optimal calcination time of 2–3 h was selected to decrease energy consumption. This duration was lower than the traditional calcination time (>4 h) utilized industrially.

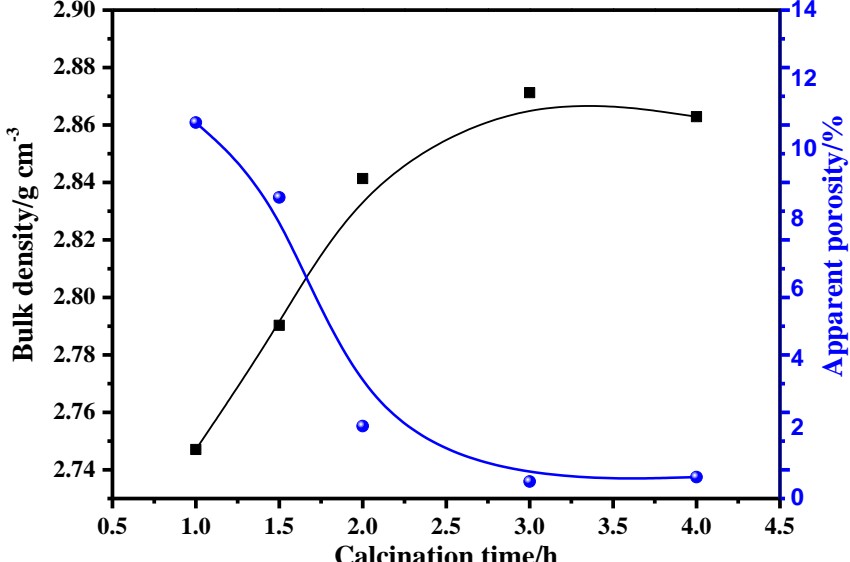

**Figure 10.** Effects of calcination time on mullite properties.

The morphology of the mullite under different calcination times is presented in Figure 11. Although the diffraction peak intensity of the mullite was invariant relative to sintering time, the small aggregated mullite seeds were separated and became a complete mullite with high aspect ratio, which was beneficial for the formation of a stable structure with increased sintering times from 1 h to 3 h. However, the separated clubbed mullite crystals accumulated again when the sintering time was controlled at 4 h, and the stable spatial skeleton structure was shrunken and destroyed. Therefore, based on these results and analysis, an optimal sintering time of 2–3 h was selected to guarantee the complete crystallization of mullite.

After systematic optimization of the process and characterization analysis, the optimal conditions were identified as follows: W, 8%; A, 0%; P, 168 MPa; T, 1650 °C; and t, 2–3 h. In accordance with the Metallurgical Industrial Standard of Synthetic Mullite (YB/T5267-2013), the refractoriness and content of the mullite phase were also measured thrice, in addition to the content of related elements, bulk density, and apparent porosity under optimal conditions. The resulting product exceeded standard requirements (Table 3).

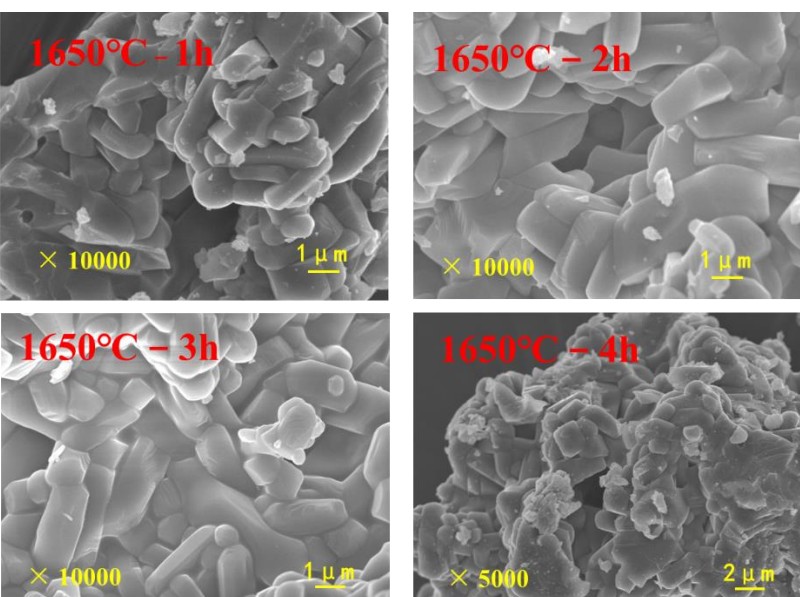

**Figure 11.** Morphology of mullite under different calcination times.

**Table 3.** Comparison of properties between experimental product and standard requirements.

| | Product Level 1 | Product Level 2 | Product Level 3 | Standard Requirements (M70-2) |
|---|---|---|---|---|
| $Al_2O_3$ (%) | 69.76 | 70.17 | 70.02 | 67–72 |
| $TiO_2$ (%) | 1.42 | 1.47 | 1.48 | $\leq 3.5$ |
| $Fe_2O_3$ (%) | 0.74 | 0.67 | 0.71 | $\leq 1.5$ |
| $Na_2O + K_2O$ (%) | 0.34 | 0.32 | 0.32 | $\leq 0.4$ |
| Bulk density ($g/cm^3$) | 2.87 | 2.92 | 2.94 | $\geq 2.75$ |
| Apparent porosity (%) | 0.37 | 0.42 | 0.41 | $\leq 5$ |
| Refractoriness (CN) | 180 | 180 | 180 | 180 |
| Mullite content (%) | 92.35 | 93.44 | 94.08 | $\geq 90$ |

### 3.4. Mechanism of Activation–Deep Desilication–Sintering Process

The existence of amorphous silicate with a lower Al/Si mass ratio decreases the A/S of HAFA, which hinders mullitization. In terms of this problem, a presumptive schematic diagram of the mechanism of mullite separation and growth was prepared (Figure 12). The Al/Si mass ratio increases with the decomposition of amorphous silicate during the hydrometallurgical process (Figure 12A–C). From Figure 12A,B, it can be concluded that the inert aluminosilicate attracts $H^+$ to form a reactive silanol structure via Al/H replacement, and the reactive silanol can be decomposed by a dilute OH- solution (Figure 12C); hence, the deep separation between amorphous silicate and mullite grains is accomplished easily, yielding a high-quality raw material. The mullite grains are grown to virgulate mullite structure, which can exhibit better properties (Figure 12C–E). Below a sintering temperature of 1200 °C, the small mullite grains (Figure 12C) vertically grow into vimineous mullite (Figure 12D), which does not exhibit improved properties. When the sintering temperature is elevated to 1650 °C, the vimineous mullite structure can be grown and melted, forming a spatial skeleton structure that can yield better properties.

$$\text{A-B: } nSiOH(O\text{-}Al(OH)_2)_3 + 9nH \rightarrow nSi(OH)_4 + 6nH_2O + 3nAl^{3+} \tag{1}$$

$$\text{B-C: } nSi(OH)_4 + 2nNaOH \rightarrow nNa_2SiO_3 + 3nH_2O \tag{2}$$

$$\text{C-D: } n\ 3Al_2O_3.2SiO_2 \rightarrow m\ 3Al_2O_3.2SiO_2\ (m < n) \tag{3}$$

$$\text{D-E: } m\ 3Al_2O_3.2SiO_2 \rightarrow x\ 3Al_2O_3.2SiO_2\ (x \ll n) \tag{4}$$

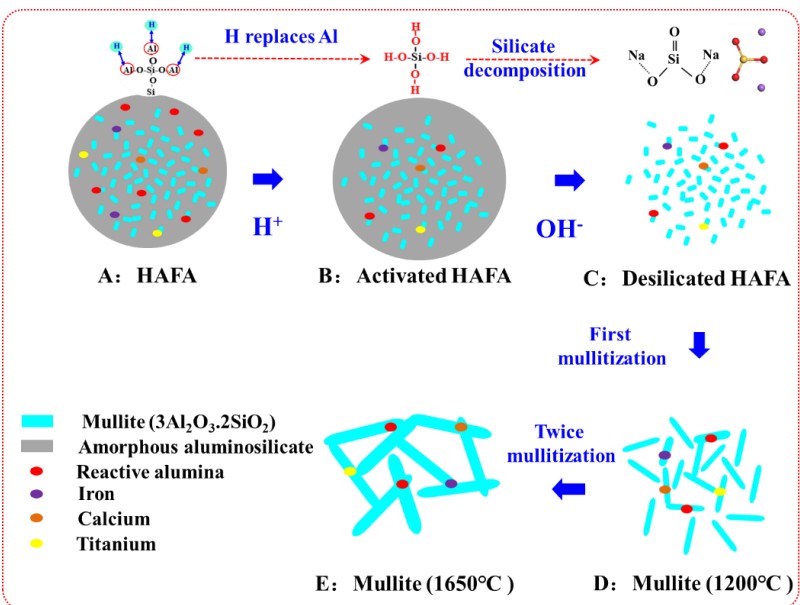

**Figure 12.** Schematic diagram of mechanism of separation and growth of mullite. (**A**,**B**): Acid activation; (**B**,**C**): Deep desilication; (**C**,**D**): Early mullitization under 1200 °C; (**D**,**E**): Final mullitization under 1650 °C.

## 4. Conclusions

Given the low A/S and poor plasticity of HAFA, an aluminosilicate activation–deep desilication–sintering synergistic method was put forward. The A/S and bulk density could be enhanced to above 2.80 and 2.85 g/cm$^3$, respectively. The mechanisms of the acid activation–deep desilication–sintering synergistic process were also investigated in detail.

(1) Mineral phases in HAFA include mullite/corundum/amorphous aluminosilicate. Crystal phases (mullite/corundum) are wrapped by an amorphous phase, which is mainly in the form of $Q^4$(3,2,1,0Al).

(2) During deep desilication, the active Al-O- in the amorphous aluminosilicate phase is decomposed and replaced by H$^+$, which helps to form an active OH site, and the amorphous phase becomes flocculent. The Al/Si ratio was increased from the original 1.17 to 2.80 through this method.

(3) During the sintering period, the mullite grains grew into rod-like structures, which helped to form a stable, intricate structure under the following optimal conditions: forming pressure, 168 MPa; moisture content, 8%; calcination temperature, 1650 °C; and calcination time, 2 h. The bulk density of the mullite was enhanced to above 2.85 g/cm$^3$, and the apparent porosity was controlled below 0.5%.

**Author Contributions:** Conceptualization, methodology, formal analysis, investigation, resources, data curation, writing—original draft preparation, J.Z.; writing—review and editing, H.L. and S.L.; visualization, J.Z.; supervision, J.Z.; project administration, H.L. and J.Z.; funding acquisition, H.L. and J.Z. All authors have read and agreed to the published version of the manuscript.

**Funding:** The work was finally supported by the National Nature Science Foundation of China (Grant No. U1810205, 52174390, 51804293), National Key Research and Development Program of China (2019YFC1904302, 2021YFC2902602), Innovation Academy for Green Manufacture of the Chinese Academy of Sciences China (IAGM2022D04).

**Conflicts of Interest:** The authors declare no conflict of interest.

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
