# Peer review of "Mechanisms of Separation and Crystal Growth of Mullite Grains during Preparation of Mullite-Based Ceramics from High Alumina Coal Fly Ash"

_processes, doi:10.3390/pr10112416_

Round 1

Reviewer 1 Report

In this manuscript, the mechanisms of separation and crystal growth of mullite grains during preparation of mullite-based ceramics from high alumina coal fly ash has been investigated, which can provide guidance for the comprehensive utilization of HAFA, so this research is important for the power plants. However, some questions should be explained in detail, so minor revision is needed.

1)    Why is the structure of Q4(3,2,1Al) decomposed by acid, but the structure of Q4(4Al) can not be decomposed?

2)    Amorphous silica can react with the NaOH solution easily, why not choose the direct desilication process?

3)    The summation of chemical composition of original HAFA is not 100%, please explain it.

4)    What is difference of preparation of mullite ceramics between this process with traditional process?

5)    From the Fig.11, the morphology is kept invariant, why does the density change a lot?

6)    How does the shape pressure and moisture content affect the properties of mullite ceramics?

7)    The impurities can affect the properties of product, and how to deal with this problem in this research?

Author Response

Thanks for the reviewer’s suggestion. According to the comments, some explanation and details are stated point to point, and the explanation is as followings, and the revised content is marked in red.

1)Why is the structure of Q4(3,2,1Al) decomposed by acid, but the structure of Q4(4Al) can not be decomposed?

Reply: Thanks for the reviewer’s comment. Q4(3,2,1Al) structure stands for the amorphous silicate, which is reactive, and the metal oxide (Fe2O3/Al2O3/CaO) can be decomposed easily in acid solution. However, Q4(4Al) stands for stable mullite phase, which is not decomposed in concentrated acid solution. Therefore, the structure of Q4(3,2,1Al) can be decomposed by acid, but the structure of Q4(4Al) can not be decomposed.

2)Amorphous silica can react with the NaOH solution easily, why not choose the direct desilication process?

Reply: Thanks for the reviewer’s comment. Though amorphous silica can react with the NaOH solution easily, only about 30%-40% amorphous silica can be dissolved in NaOH solution because part of amorphous phase (Q4(3,2,1Al)) is stable, and the existence of Al can lead to serious side reaction of zeolite, which can prevent the deep desilication. If direct desilication process is adopted, more Na-zeolite is formed easily, and the existence of zeolite decreases the refractoriness of mullite ceramics. Therefore, the acid activation-deep desilication process is promoted, and the direct desilication process can not meet the demand of product.

3)The summation of chemical composition of original HAFA is not 100%, please explain it.

Reply: Thanks for the reviewer’s comment. According to the suggestion, the composition of other elements(K/S/C and so on) occupies little, so this part is not exhibited in table, and the table has been edited, which make the summation of chemical composition of original HAFA reach 100%.

4)What is difference of preparation of mullite ceramics between this process with traditional process?

Reply: Thanks for the reviewer’s comment. Traditionally, HAFA (Al/Si mass ratio≈1.0) and bauxite are mixed to elevate the Al/Si mass ratio(Al/Si mass ratio≈2.7), and the mullite ceramics((Al/Si mass ratio≈2.55))is prepared by sintering process under the conditions: 1650℃, >12h. In contrast, only HAFA is adopted without addition of other Al resources in this process, and the sintering time is only controlled bellow 3h, which can decrease the consumption of Al resources and energy.

5)From the Fig.11, the morphology is kept invariant, why does the density change a lot?

Reply: Thanks for the reviewer’s comment. Under the temperature 1650℃, the mullite grains grows fast, so the morphology exhibit invariant, but there are differences under different sintering time. When sintering time is below 2h, it is seen that some space can not be grown and gathered, so the bulk density is lower, and apparent porosity is higher. When the sintering time exceeds 2h, the hole disappeared, and the clubbed mullite are gathered to form a dense body. Therefore, the change of bulk density and apparent porosity of mullite is consistent with the change of mullite morphology.

6)How does the shape pressure and moisture content affect the properties of mullite ceramics?

Reply: Thanks for the reviewer’s comment. DHAFA has a poor plasticity, which can not be sintered directly. The shaping pressure and moisture content in the forming process play a positive role on the elevation of plasticity, which can help to form a dense body, and the mullite grains are gathered better during thre sintering process, and bulk density can be elevated to a high level.

7)The impurities can affect the properties of product, and how to deal with this problem in this research?

Reply: Thanks for the reviewer’s comment. The impurities of Na/Ca/K/Mg can decrease the refractoriness of mullite product. Ti can decrease the creep property. Fe affects the color of product, and prevent the sintering of mullite grains. Therefore, the acid treatment is adopted, which can not only activate the amorphous silicate but also remove the impurities of Na/Ca/K/Mg/Fe, so the negative effects are removed. Ti is in the form of TiAlSiO4, which belong to a kind of refractory, so Ti in HAFA can not affect the creep property.

Reviewer 2 Report

Coal fly ash is one of the largest industrial solid wastes in the world, and its utilization is a thorny problem that people are faced with. This manuscript aims to prepare mullite-based ceramics using the high alumina coal fly ash. A hydrometallurgy pyrometallurgy synergistic process was proposed to resolve the lower Al2O3/SiO2 mass ratio (Al/Si) and lower degree of crystallinity in this work. This work is interesting and meaningful. The results were clearly organized and the main conclusions are supported by the data. This work can be accepted after addressing the following flaws:

1) During the preparation of mullite ceramics, why is the Al2O3/SiO2 mass ratio elevated?

2) How does the Si-O-Al coordination structure affect the reactivity of amorphous silicate?

3) The Al2O3 can be leached out by acid process, which can decrease the Al/Si mass ratio, why not adopt desilication process directly?

4) During this process, acid is adopted to activate HAFA, but how to deal with the acid solution?

5) How to determine the conditions of acid activation and desilication process?

6) During the desilication process, Na-zeolite can be easily formed, which can decrease the refractoriness of mullite ceramics. How to control this problem?

7)TiO2 play a negative role on the creep property of refractory materials, why not remove the TiO2?

Author Response

Thanks for the reviewer’s suggestion. According to the comments, some explanation and details are stated point to point, and the explanation is as followings, and the revised content is marked in red.

1)During the preparation of mullite ceramics, why is the Al2O3/SiO2 mass ratio elevated?

Reply: Thanks for the reviewer’s comment. Mullite/corundum/amorphous silicates are the main phases in HAFA, and Al/Si mass ratio of HAFA is about 1.0 due to the existence of amorphous silicates, which can decrease the sintering properties. However, Al/Si mass ratio in standard mullite product is 2.55, so Al2O3/SiO2 mass ratio should be elevated to be above 2.55, which can promote the formation of mullite. Therefore, Al2O3/SiO2 mass ratio must be elevated.

2)How does the Si-O-Al coordination structure affect the reactivity of amorphous silicate?

Reply: Thanks for the reviewer’s comment. The Si-O-Al coordination structures include Q4(3Al)/Q4(2Al)/ Q4(1Al), and the Al-O- bonds affect the reactivity of Si-O-. When the Al-O- bonds around Si-O- are increased, the reactivity of Si-O- is decreased. Simultaneously, the existence of Al can lead to serious side reaction of zeolite, which can prevent the deep desilication. Therefore, the Al-O- should be removed to increase the reactivity of Si-O-.

3)The Al2O3 can be leached out by acid process, which can decrease the Al/Si mass ratio, why not adopt desilication process directly?

Reply: Thanks for the reviewer’s comment. Though the Al/Si mass ratio is decreased from 1.2 to 1.0 or so due to removal of Al by acid treatment, the Al/Si mass ratio can be increased from 1.0 to above 2.7 due to the high reactivity of amorphous silica by acid activation-deep desilication process. If direct desilication process is adopted, more Na-zeolite is formed easily, and the existence of zeolite decreases the refractoriness of mullite ceramics. Therefore, the acid activation-deep desilication process is promoted, and the direct desilication process can not meet the demand of product.

4)During this process, acid is adopted to activate HAFA, but how to deal with the acid solution?

Reply: Thanks for the reviewer’s comment. Al3+/Ca2+/Fe3+/H+/Cl- are the main ions in acid solution after the acid activation process, and acid solution is recycled to activate the amorphous silicate due to the high concentrated H+, simultaneously, the Al3+/Ca2+/Fe3+ are enriched. When the concentration of Al3+reach above 20g/L, it is adopted to prepare polymeric aluminium (PAC), which can help to accomplish the comprehensive utilization of HAFA.

5)How to determine the conditions of acid activation and desilication process?

Reply: Thanks for the reviewer’s comment. The conditions in this research have been optimized in previous study.

Related reference: Jianbo Zhang, Shaopeng Li, Huiquan Li, Qisheng Wu, Xinguo Xi, Zhanbing Li. Preparation of mullite ceramic from high-alumina coal fly ash through mechanical-chemical synergistic activation. Ceramics International, 2017, 43: 6532-6541.

6)During the desilication process, Na-zeolite can be easily formed, which can decrease the refractoriness of mullite ceramics. How to control this problem?

Reply: Thanks for the reviewer’s comment. When direct desilication process is adopted, Na-zeolite can be easily formed due to existence of Na+/Al3+/SiO32-/OH-. However, reactive Al in amorphous phase is leached into acid solution under the attack of H+, so little Al is leached into desilicated solution to form AlO2-. Therefore, Na-zeolite can not be easily formed due to lack of Al, and Na content is controlled below 0.4%, which can not affect the refractoriness of mullite ceramics.

7)TiO2 play a negative role on the creep property of refractory materials, why not remove the TiO2?

Reply: Thanks for the reviewer’s comment. Ti is in the form of TiAlSiO4, which can not be decomposed by concentrated acid. At the same time, TiAlSiO4 belong to a kind of refractory, so Ti in HAFA can not affect the creep property.

Reviewer 3 Report

I think that the topic of the article “Mechanisms of separation and crystal growth of mullite grains during preparation of mullite-based ceramics from high alumina coal fly ash” is very actual. In my opinion, the article can be published in the form in which it is presented.

Author Response

Thanks for the reviewer’s recognition.